# Rheological Behavior of DNP/HMX Melt-Cast Explosives with Bimodal and Trimodal Particle-Size Distributions

**DOI:** 10.3390/polym15061446

**Published:** 2023-03-14

**Authors:** Hanfei Xie, Xiangrong Zhang, Tao Jiang, Yingzhong Zhu, Lin Zhou

**Affiliations:** 1State Key Laboratory of Explosion Science and Technology, Beijing Institute of Technology, Beijing 100081, China; 2Chongqing Hongyu Precision Industry Group Co., Ltd., Chongqing 402760, China

**Keywords:** melt-cast explosive suspension, apparent viscosity, bimodal and trimodal particle-size distribution, shear rate, Krieger–Dougherty relation

## Abstract

As a matrix for melt-cast explosives, 3,4-dinitropyrazole (DNP) is a promising alternative to 2,4,6-trinitrotoluene (TNT). However, the viscosity of molten DNP is considerably greater compared with that of TNT, thus, requiring the viscosity of DNP-based melt-cast explosive suspensions to be minimized. In this paper, the apparent viscosity of a DNP/HMX (cyclotetramethylenetetranitramine) melt-cast explosive suspension is measured using a Haake Mars III rheometer. Both bimodal and trimodal particle-size distributions are used to minimize the viscosity of this explosive suspension. First, the optimal diameter ratio and mass ratio (two crucial process parameters) between coarse and fine particles are obtained from the bimodal particle-size distribution. Second, based on the optimal diameter ratio and mass ratio, trimodal particle-size distributions are used to further minimize the apparent viscosity of the DNP/HMX melt-cast explosive suspension. Finally, for either the bimodal or trimodal particle-size distribution, if the original data between the apparent viscosity and solid content are normalized, the resultant plot of the relative viscosity versus reduced solid content collapses to a single curve, and the effect of the shear rate on this curve is further investigated.

## 1. Introduction

Although additive manufacturing (3D printing) technologies are emerging in the energetic materials area [1], traditional formatting approvals for energetic materials, especially for melt-cast explosives still dominate. Improving the pourability of melt-cast explosives has at least a twofold significance. First, good pourability aids melt-cast explosive suspensions to flow completely into a mold or projectile casing, which helps eliminate voids or other defects in the final explosive charges. Since the voids and other defects can form potential hot spots in the explosive charges upon experiencing external shock or mechanical stimuli [2], good pourability can improve the safety of explosive charges. Second, good pourability (low viscosity) also helps to achieve a high solid content in melt-cast explosives [3], thus, improving the performance of explosive charges. Generally, good pourability can be accomplished by minimizing the viscosity of melt-cast explosive suspensions. Furthermore, the viscosity depends on many factors, such as the temperature, chemical additives, shear rate, particle morphology, and size distribution [4,5,6,7,8,9].

As one of the most important factors affecting viscosity, the particle-size distribution plays a key role in minimizing the suspension viscosity [10,11,12]. Mixing particles of different sizes is a common way to minimize viscosities because it gives a broader particle-size distribution so that finer particles may fit into the spaces between coarser packed particles [13,14,15].

Therefore, a bimodal particle-size distribution is usually better than a unimodal distribution [16,17], and similarly a trimodal or multimodal distribution is better than a bimodal distribution. However, in practical industrial production or even in laboratory investigations, a balance between quality and efficiency is usually considered when minimizing viscosity so that bimodal distributions are more frequently used compared with other distributions [9,18,19]. Furthermore, for bimodal particle-size distributions, the minimization of viscosity may be affected by two crucial process parameters [20,21,22,23]: (i) the ratio λ of the diameter ($d_{50}$ in the present study) of coarse particles to that of fine particles and (ii) the ratio ζ of the mass of coarse particles to that of fine particles.

Based on published values [9,16,20,22,24,25,26,27], λ is usually in the range of 1–10, while ζ is usually in the range of 0.1–4. Generally, if λ is specified, an optimal ζ exists to minimize the viscosity of suspensions under a certain high solid content; vice versa, there exists an optimal λ if ζ is specified. However, such an optimal λ or ζ is generally determined on the basis of limited experimental comparison schemes, so they are usually local optima rather than global optima. Furthermore, if the concepts of λ and ζ are extended to trimodal or multimodal particle-size distributions, it is even harder to experimentally determine their global optimum.

As a new matrix ingredient in melt-cast explosives, DNP is a promising alternative to TNT [28,29]. The melting point and mechanical sensitivity of DNP are comparable to those of TNT [30], while the performance (e.g., the detonation velocity or detonation pressure) of DNP is much better than that of TNT [31,32]. Conversely, the viscosity of DNP is about 80% greater than that of TNT [33], which is disadvantageous for obtaining a suspension with high solid content. Therefore, compared with TNT-based melt-cast explosive suspensions, the minimal-viscosity requirement of DNP-based melt-cast explosive suspensions is more stringent.

Although Zhou has investigated the viscosity minimization of DNP/HMX melt-cast explosive suspensions with bimodal particle-size distributions [33], he considered only a solid content of 60 wt.% and a diameter ratio λ=35. Since the viscosity of DNP is much greater than that of TNT, it is essential to investigate the rheological behavior of DNP-based melt-cast explosive suspensions at different solid-content and diameter ratios. Moreover, the particle-size distribution should not be limited to bimodal distributions; trimodal distributions should also be investigated and compared.

We, thus, present herein a detailed study of the rheological behavior of DNP/HMX melt-cast explosive suspensions with bimodal and trimodal particle-size distributions. The viscosity of the suspensions is measured using a Haake rheometer. Bimodal particle-size distributions are first investigated during the process of viscosity minimization of the DNP/HMX melt-cast explosive suspensions, then trimodal particle-size distributions are investigated to further minimize the viscosity of the suspensions, and finally the dependence of relative viscosity on reduced solid content is analyzed for both bimodal and trimodal particle-size distributions.

## 2. Experiments

### 2.1. Materials

Both the DNP and HMX used in this study were supplied by the Yinguang Chemical Industry Group Co., Ltd. (Beiyin, Gansu, China) with purities of 99.1% and 99.4%, respectively. As shown in Figure 1, DNP powder is yellow, whereas the crux HMX particles are white. Figure 1b,c show the photographs of HMX particles (sample $S_{3}$) taken by camera and by scanning electron microscope (SEM), respectively. Moreover, since the HMX particles used in the formulation of explosives generally measure 1–100μm, six HMX particle-size distributions (labeled $S_{1}$–$S_{6}$ in Figure 2) were collected and used to investigate how bimodal and trimodal particle-size distributions affect the viscosity minimization of DNP/HMX explosive suspensions. The mean particle size $d_{50}$ of samples $S_{1}$–$S_{6}$ was 5.6, 9.2, 38.8, 162.7, 269.6, and 366.6 μm, respectively.

### 2.2. Schemes of Particle Gradation

As mentioned in Section 1, a bimodal particle-size distribution is commonly used to minimize the viscosity of explosive suspensions. Thus, we first investigate how a bimodal particle-size distribution affects the apparent viscosity of DNP/HMX melt-cast explosive suspensions. Based on the optimal bimodal particle-size distribution determined for λ and ζ, we then use a trimodal particle-size distribution to further reduce the apparent viscosity of the DNP/HMX melt-cast explosive suspensions.

Based on the six samples $S_{1}$–$S_{6}$, there are 15 (given by the combination C(6,2)) possible values for λ. For a practical bimodal particle-size distribution for melt-cast explosives, however, the order of magnitude of the mean particle size is $10\;\mu m\leq d_{50}\leq 100\;\mu$m [9,33,34,35,36]. Samples with an order of magnitude of the mean particle size of $d_{50}>100\;\mu$m are seldom used because of serious sedimentation [9]. Furthermore, samples with an order magnitude of the mean particle size $d_{50}<10\;\mu$m are avoided because they lead to a large viscosity [9].

Moreover, the typical published values for λ are usually in the range of 1–10 [16,22,25,27]. Therefore, only samples $S_{3}$–$S_{6}$ with three combinations ($S_{3}$ and $S_{4}$, $S_{3}$ and $S_{5}$, and $S_{3}$ and $S_{6}$) were used to investigate the viscosity minimization with bimodal particle-size distribution. The values of λ for the three combinations are about 4, 7, and 9 (rounded to integer values), respectively. Compared to λ, the published values for ζ are usually in the range of 0.1–4 [9,20,24,26]. This study compares only five values for ζ: 0.25, 0.5, 1, 2, and 4. Combining the values of λ with those of ζ gives a total of 15 schemes for particle gradation, which are listed in Table 1, where $\lambda_{43}$ is the diameter ($d_{50}$) ratio of sample $S_{4}$ to that of sample $S_{3}$, and $\zeta_{43}$ is the mass ratio of sample $S_{4}$ to that of sample $S_{3}$. Likewise, $\lambda_{53}$, $\zeta_{53}$, $\lambda_{63}$, and $\zeta_{63}$ have similar physical meanings.

After investigating the viscosity minimization with bimodal particle-size distribution, viscosity minimization is further investigated with trimodal particle-size distribution. This time, not only are samples $S_{3}$–$S_{6}$ used but also samples $S_{1}$ and $S_{2}$. This issue is further discussed in Section 3.2.

### 2.3. Viscosity Measurement

The apparent viscosity of the DNP/HMX melt-cast explosive suspensions was measured using an R/S Haake Mars III rheometer (Thermo Fisher Scientific Inc., Waltham, MA, USA). The measuring element has a Couette geometry with a gap of 2.0 mm. Furthermore, the inner and outer radii of the Couette geometry were 8.5 and 10.5 mm, respectively. Therefore, the assumption of a constant shear rate in the gap was respected, the particle migration [37,38,39] was assumed to be negligible, and the material was assumed to be well-mixed.

Moreover, wall slip is commonly observed in rheometry suspensions, and such phenomena are generally associated with many factors, such as low shear rates and smooth walls of the measuring geometries [40,41]. However, the specific analysis of the wall slip phenomenon will not be detailed in this study. An explosion-proof cover was installed outside the original measuring element of the rheometer. As shown in Figure 3, although this cover installs quite simply, it can resist a blast wave if a sample accidentally explodes.

The procedure to measure the apparent viscosity has the following main steps:1.In a double-jacketed stainless-steel kettle, DNP powders are first melted at 100 ${}^{\circ }$C (where an oil bath was used due to the risk of explosive samples), following which HMX particles were added incrementally to the molten DNP.2.The DNP/HMX mixture was stirred for 15 min at a rate of 500 rev/min to ensure uniform mixing and to eliminate solid agglomerates.3.Uniform DNP/HMX suspensions were poured into the measuring element of the rheometer, and the apparent viscosities of the explosive suspensions were measured at shear rates in the range of 1–1000 $s^{-1}$.

## 3. Results and Discussion

### 3.1. Apparent Viscosity with Bimodal Size Distribution

To determine the optimal values λ and ζ of the DNP/HMX melt-cast explosive suspensions with a bimodal particle-size distribution, two optional methods were used. One method is to investigate how the shear rate affects the apparent viscosity of these explosive suspensions for a given solid content, and another method is to investigate how the solid content affects the apparent viscosity at a given shear rate.

#### 3.1.1. Shear Thinning Viscous Behavior

Melt-cast explosive suspensions are solid–liquid two-phase mixtures. Generally, the liquid phase exhibits a Newtonian viscous behavior [3,42,43]. However, when the solid phase is dispersed in the liquid phase, the resultant melt-cast explosive suspensions usually exhibit a non-Newtonian (shear thinning) viscous behavior [6,7,8]. Furthermore, the relationship between the apparent viscosity of these suspensions and shear rates approximately follows a power law [9,44].

Figure 4 shows that, for any scheme listed in Table 2, the apparent viscosity $\eta_{a}$ of the DNP/HMX melt-cast explosive suspension decreases exponentially with the shear rate $\dot{\gamma }$. The slopes between log ($\eta_{a}$) and log ($\dot{\gamma }$) are negative, demonstrating that the DNP/HMX melt-cast explosive suspension presents a non-Newtonian (shear thinning) viscous behavior. Moreover, for any diameter ratio λ ($\lambda_{43}\approx 4$, $\lambda_{53}\approx 7$, $\lambda_{63}\approx 9$), the best parameter for mass ratio ζ is the same for all ζ ($\zeta_{43}=\zeta_{53}=\zeta_{63}=2$). However, compared with the cases of $\lambda_{43}$ and $\lambda_{63}$, the case of $\lambda_{53}$ corresponds to the minimum viscosity of the DNP/HMX melt-cast explosive suspensions. This is further demonstrated in Table 2, when samples $S_{5}$ ($d_{50}=269.6\;\mu m$) and $S_{3}$ ($d_{50}=38.8\;\mu m$, $\lambda_{53}\approx 7$) are mixed together with a mass ratio $\zeta_{53}=2$. The viscosity of this DNP/HMX explosive suspension is, thus, minimized.

#### 3.1.2. Effects of Solid Content

Compared with the shear thinning behavior, the apparent viscosity $\eta_{a}$ of the DNP/HMX melt-cast explosive suspensions increases with the solid content ϕ. However, whether based on $\eta_{a}$-$\dot{\gamma }$ curves or on $\eta_{a}$-ϕ curves, the best parameters for λ and ζ are the same; i.e., when samples $S_{5}$ ($d_{50}=269.6\;\mu$m) and $S_{3}$ ($d_{50}=38.8\;\mu$m; $\lambda_{53}\approx 7$) are mixed together with mass ratio $\zeta_{53}=2$, the apparent viscosity of the DNP/HMX melt-cast explosive suspensions is minimized. Further details appear in Figure 5 and Table 3.

### 3.2. Apparent Viscosity with Trimodal Size Distribution

Compared with the bimodal particle-size distribution, trimodal particle-size distribution produces many more combinations, with different diameter ratios combined with different mass ratios. Therefore, it is much more difficult to experimentally determine the optimal values of λ and ζ with trimodal particle-size distributions. One way to reduce this difficulty is to further minimize the viscosity based on the optimal values of λ and ζ obtained from bimodal particle-size distributions.

However, the concepts of λ and ζ must be extended for trimodal particle-size distributions. For bimodal particle-size distributions, λ represents the diameter ($d_{50}$) ratio of coarse particles to fine particles, and ζ represents the mass ratio of coarse particles to fine particles. In contrast, for trimodal particle-size distributions, λ and ζ represent the diameter and mass ratios for coarse, medium, and fine particles.

Furthermore, for trimodal particle-size distributions, particles with an order of magnitude diameter $d_{50}\in 1$, 10, and 100μm are defined as fine, medium, and coarse particles, respectively. Based on this definition, the present samples $S_{1}$ and $S_{2}$ contain fine particles, $S_{3}$ contains medium particles, and $S_{4}$, $S_{5}$, and $S_{6}$ contain coarse particles. For bimodal particle-size distributions, sample $S_{3}$ contains fine particles (see Section 3.1).

Moreover, similar to the value of λ with bimodal particle-size distributions, the value of λ with trimodal particle-size distributions is also rounded to the nearest integer. For example, $\lambda_{63}$ is rounded to 9, $\lambda_{31}$ is rounded to 7, and $\lambda_{631}$ is rounded to 63:7:1. In contrast, ζ is not rounded to integer values for either bimodal or trimodal particle-size distributions.

#### 3.2.1. Fixed Diameter Ratio

Given that the optimal diameter ratio for bimodal particle-size distributions has been experimentally determined ($\lambda_{53}$), we assume that the optimal diameter ratio remains unchanged for trimodal particle-size distributions (i.e., the diameter ratio between coarse and medium particles is the same as that between medium and fine particles).

Furthermore, for a trimodal particle-size distribution, $\lambda_{53}$ is the diameter ratio between coarse (sample $S_{5}$) and medium (sample $S_{3}$) particles. The fine particles must then be sample $S_{1}$, not $S_{2}$, since the diameter ratio $\lambda_{31}$ is rounded to 7 and equal to $\lambda_{53}$, whereas $\lambda_{32}$ is rounded to 4. Therefore, the diameter ratio for the trimodal particle-size distribution is denoted $\lambda_{531}$ and is rounded to 49:7:1.

Given the diameter ratio $\lambda_{531}$, we investigated how different values of the mass ratio $\zeta_{531}$ affect the apparent viscosity of the DNP/HMX melt-cast explosive suspensions. As shown in Figure 6, the mass ratio $\zeta_{53}=2$ equals the optimal value for bimodal particle-size distributions, whereas the mass ratio $\zeta_{31}$ varies in the range of 1.25–5, as listed in Table 4. In any case, the apparent viscosity of this explosive suspension is less than that (12.7 Pa·s; see Figure 4 and Table 2) of a bimodal particle-size distribution with the optimal diameter and mass ratios ($\lambda_{53}\approx 7$, $\zeta_{53}=2$).

However, the viscosity is not minimized with the mass ratio $\zeta_{53}=\zeta_{31}=2$ (or equivalently $\zeta_{531}=2:1:0.5$), although, in this case, the apparent viscosity (10.33 Pa·s) is about 19% less than the minimum viscosity (12.7 Pa·s) of bimodal particle-size distributions. Instead, the minimum viscosity (9.19 Pa·s) is obtained with the mass ratio $\zeta_{531}=2:1:0.3$, which is about 28% less than the minimum viscosity of the bimodal particle-size distributions.

#### 3.2.2. Fixed Mass Ratio

As in the above discussion for fixed diameter ratio, we again assumed that, for trimodal particle-size distributions, the mass ratio between coarse and medium particles is the same as that between medium and fine particles (i.e., $\zeta_{\mathrm{cm}}=\zeta_{\mathrm{mf}}=2$, or equivalently $\zeta_{\mathrm{cmf}}=2:1:0.5$, where the subscripts “c,” “m,” and “f” indicate coarse, medium, and fine particles, respectively).

Following the above definition of coarse, medium, and fine particles gives six combinations of diameter ratios for trimodal particle-size distributions: $\lambda_{431}$, $\lambda_{531}$, $\lambda_{631}$, $\lambda_{432}$, $\lambda_{532}$, and $\lambda_{632}$, as listed in Table 5. As shown in Figure 7, for any of the six diameter ratios, the apparent viscosity of this explosive suspension is less than that of the bimodal particle-size distribution with the optimal diameter and mass ratios ($\lambda_{53}\approx 7$, $\zeta_{53}=2$).

However, as with the case in Section 3.2.1, the viscosity is not minimized with the diameter ratio $\lambda_{53}=\lambda_{31}\approx 7$ (or equivalently $\lambda_{531}\approx 49:7:1$), instead, the minimum viscosity (9.58 Pa·s) is obtained with the diameter ratio $\lambda_{631}$, which is about 25% less than the minimum viscosity of bimodal particle-size distributions.

### 3.3. Relative Viscosity Related to Reduced Solid Content

As shown in Section 3.1, the experimental data relating to the apparent viscosity $\eta_{a}$ and to the solid content ϕ are characterized by several curves. However, these curves may collapse to a single curve if the solid content is normalized by the maximum solid content $\phi_{m}$. The normalized solid content $\phi /\phi_{m}$ is called the “reduced solid content”. In the literature, the relative viscosity $\eta_{r}$ is usually plotted as a function of the reduced solid content [10,14]: $\eta_{r}=f$$\left(\phi /\phi_{m}\right)$, where $\eta_{r}\equiv \eta_{a}/\eta_{0}$ and $\eta_{0}$ is the Newtonian viscosity of the liquid phase.

For general applications in suspension rheology, the equation proposed by Krieger and Dougherty is a commonly used semi-empirical relation between relative viscosity $\eta_{r}$ and reduced solid content $\phi /\phi_{m}$ and is given by [45,46]
(1)$$\eta_{r}={\left(1-\phi /\phi_{m}\right)}^{-\alpha },$$
where α characterizes the divergence when ϕ approaches $\phi_{m}$. For rigid spheres, α is usually between one and two [46]. However, for the present non-spherical HMX particles, α may well exceed the above range.

Moreover, when the parameter is determined by the best fit of Equation (Equation 1) to the experimental data, the maximum solid content $\phi_{m}$ is obtained simultaneously. Simply plotting ${\left(1/\eta_{r}\right)}^{\alpha }$ against ϕ yields a straight line so that $\phi_{m}$ is the *x* intercept when ${\left(1/\eta_{r}\right)}^{\alpha }\to 0$ [47]. This straight line must pass through the point (0,1) because, when no particles are dispersed in the liquid phase (ϕ→0), the apparent viscosity $\eta_{a}$ of the suspensions equals the Newtonian viscosity $\eta_{0}$ of the liquid phase (i.e., $\eta_{r}\to 1$).

Combining the apparent viscosity data (Figure 5) with the Newtonian viscosity of the liquid phase ($\eta_{0}$ for DNP is 16.4 mPa·s at 100 ${}^{\circ }$C [33]), the parameter α is determined to be 3.94, whereas the maximum solid content $\phi_{m}$ varies with both λ and ζ (i.e., $\phi_{m}=\phi_{m}\left(\lambda ,\zeta \right)$) [46]. This is physically reasonable. Generally, $\phi_{m}$ is attained when neighboring particles are in permanent contact so that the particles are unable to move past one another, in which case the suspension viscosity becomes infinite [48]. Assuming a fixed particle shape, $\phi_{m}$ is then approximately characterized by λ and ζ.

As shown by Table 6, the value of $\phi_{m}$ varies within the range of 75.1–77.7 wt.%. As part of the background, the apparent viscosities at the solid content of 65 wt.% are also listed in Table 6. Overall, $\phi_{m}$ decreases with viscosity, and the largest value of $\phi_{m}$ corresponds to the optimal diameter ratio ($\lambda_{53}\approx 7$) and mass ratio ($\zeta_{53}=2$) for the minimization of viscosity of the DNP/HMX melt-cast explosive suspensions. These facts demonstrate that the values of $\phi_{m}$ obtained are qualitatively correct.

Given the values of λ and ζ, the respective values of $\phi_{m}$ are substituted into Equation (Equation 1). The experimental relative viscosity $\eta_{r}$ correlated with reduced solid content $\phi /\phi_{m}$ is then plotted in Figure 8. Compared with the original $\eta_{a}$-ϕ data scattered on several curves (Figure 5), the data $\eta_{r}$-$\phi /\phi_{m}$ collapse to a single curve, demonstrating that Equation (Equation 1) can be used to characterize the rheological behavior of the DNP/HMX melt-cast explosive suspensions.

#### 3.3.1. Dependence of α on the Shear Rate

Based on Figure 8, it appears that the $\eta_{r}$-$\phi /\phi_{m}$ relation can be uniquely described by Equation (Equation 1) provided that the exponent α=3.94 with the known maximum solid content $\phi_{m}\left(\lambda ,\zeta \right)$. However, the experimental data used to calibrate the parameter correspond to a shear rate of 1 $s^{-1}$. Moreover, since the DNP/HMX melt-cast explosive suspensions present a non-Newtonian (shear thinning) viscous behavior (Figure 4), the relative viscosity of this explosive suspension must be a function of the shear rate, i.e., $\eta_{r}=\eta_{r}\left(\dot{\gamma }\right)$.

Therefore, the right-hand side of Equation (Equation 1) must also be a function of the shear rate (i.e., either $\phi_{m}=\phi_{m}\left(\lambda ,\zeta ,\dot{\gamma }\right)$ or $\alpha =\alpha (\dot{\gamma })$). However, as mentioned above, the maximum solid content $\phi_{m}$ corresponds to the limiting case and is dominated by the particle shape and particle-size distribution. On this account, the value of $\phi_{m}$ is assumed to have physically nothing to do with the shear rate. Consequently, the exponent α must be a function of the shear rate (i.e., $\alpha =\alpha (\dot{\gamma })$).

Following the same fitting method for parameter α as used for the shear rate of 1 $s^{-1}$, the different values of α were determined for three other typical shear rates (10, 100, and 1000 $s^{-1}$), which provided 3.33, 2.73, and 2.01, respectively. Furthermore, the α-$\dot{\gamma }$ data can be well fit by
(2)$$\alpha =-0.278\ln\left(\frac{\dot{\gamma }}{\dot{\gamma_{0}}}\right)+3.96\;\left(1\;s^{-1}\leq \dot{\gamma }\leq 1000\;s^{-1}\right),$$
where $\dot{\gamma_{0}}$ is the reference shear rate (1 $s^{-1}$). Similarly, the α-$\ln\left(\dot{\gamma }\right)$ relation for Inconel feedstocks was also investigated [49]. On the other hand, the value of $\phi_{m}$ is the same as listed in Table 6, regardless of whether the shear rate is high or low. The resultant data $\eta_{r}$-$\phi /\phi_{m}$ for the three typical shear rates are plotted in Figure 9, Figure 10 and Figure 11. As expected, for a given shear rate, these data also collapse to a single curve described by Equation (Equation 1).

Compared with the two cases with low shear rates $\dot{\gamma }=1$ and $10\;s^{-1}$ (Figure 8 and Figure 9), the experimental data for the two cases with high shear rates $\dot{\gamma }=100$ and $1000\;s^{-1}$ (Figure 10 and Figure 11) deviate more from their respective fits. This may be because the data between log ($\eta_{a}$) and log ($\dot{\gamma }$) deviate from a linear relationship at high shear rates (see Figure 4).

#### 3.3.2. Effects of Trimodal Distribution on $\phi_{m}$

The experimental data shown in Figure 8, Figure 9, Figure 10 and Figure 11 were obtained from the rheological behavior of the DNP/HMX melt-cast explosive suspensions with bimodal particle-size distributions. These $\eta_{r}$-$\phi /\phi_{m}$ data are well described by Equation (Equation 1). However, for trimodal particle-size distributions, the applicability of Equation (Equation 1) needs further investigation. We assumed that the mass ratio $\zeta_{\mathrm{cmf}}$ between coarse, medium, and fine particles is the same as the optimal value shown in Figure 6 (i.e., $\zeta_{\mathrm{cmf}}=2:1:0.3$) and considered all six combinations of diameter ratios ($\zeta_{431}$, $\zeta_{531}$, $\zeta_{631}$, $\zeta_{432}$, $\zeta_{532}$, and $\zeta_{632}$) as listed in Table 7. The corresponding $\eta_{a}$-ϕ rheological data at the four typical shear rates (1, 10, 100, and 1000 $s^{-1}$) were then measured.

Following the same fitting method for the parameter $\phi_{m}$ as used for bimodal particle-size distributions, the values of $\phi_{m}$ for trimodal particle-size distributions were obtained and are shown in Figure 12. $\phi_{m}$ for trimodal particle-size distributions is greater than $\phi_{m}$ for bimodal particle-size distributions. The corresponding $\eta_{r}$-$\phi /\phi_{m}$ data are plotted in Figure 13, Figure 14, Figure 15 and Figure 16. Not surprisingly, for a given shear rate, all $\eta_{r}$-$\phi /\phi_{m}$ data for DNP/HMX melt-cast explosive suspensions with trimodal particle-size distributions also collapse to a single curve and are uniquely described by Equation (Equation 1).

## 4. Conclusions

The apparent viscosities of the DNP/HMX melt-cast explosive suspensions with both bimodal and trimodal particle-size distributions were measured using an R/S Haake Mars III rheometer. As a matrix ingredient of melt-cast explosive suspensions, the molten DNP presented a Newtonian viscous behavior. However, when HMX particles (solid phase) were dispersed in molten DNP (liquid phase), the resultant DNP/HMX explosive suspensions presented a non-Newtonian (shear thinning) viscous behavior. The viscosity of the DNP/HMX melt-cast explosive suspensions was minimized by both bimodal and trimodal particle gradation for six HMX samples ($S_{1}$–$S_{6}$) with particle diameters spanning three orders of magnitude (1, 10, and 100μm).

For bimodal particle-size distributions, only four samples ($S_{3}$–$S_{6}$) with particle diameters spanning two orders of magnitude (10 and 100μm) were investigated. The diameter ratio λ and mass ratio ζ (two crucial process parameters) were used to characterize the bimodal particle-size distribution. Three diameter ratios (about 4, 7, and 9) and five mass ratios (0.25, 0.5, 1.0, 2.0, and 4.0) were compared. For the diameter ratios $\lambda_{43}$, $\lambda_{53}$, or $\lambda_{63}$, the best mass ratios $\zeta_{43}$, $\zeta_{53}$, or $\zeta_{63}$ were all equal to 2. However, compared with the diameter ratios $\lambda_{43}$ and $\lambda_{63}$, the diameter ratio $\lambda_{53}$ (about 7) combined with the mass ratio $\zeta_{53}=2$ produced the minimum viscosity of the DNP/HMX melt-cast explosive suspension.

Such an optimal combination of λ and ζ was determined on the basis of $\eta_{a}-\dot{\gamma }$ or $\eta_{a}-\phi$ data. For trimodal particle-size distributions, coarse ($S_{4}$–$S_{6}$), medium ($S_{3}$), and fine ($S_{1}$ and $S_{2}$) particles (with order of magnitude diameters of 100, 10, and 1μm) were all used to further minimize the viscosity of the DNP/HMX melt-cast explosive suspensions. Based on the optimal values of λ and ζ obtained for a bimodal particle-size distribution, two typical particle-gradation schemes for trimodal distribution were compared.

One was the fixed-diameter ratio scheme ($\lambda_{\mathrm{cm}}=\lambda_{\mathrm{mf}}$), and the other was the fixed-mass ratio scheme ($\zeta_{\mathrm{cm}}=\zeta_{\mathrm{mf}}$). For either scheme, the apparent viscosity was less than that obtained from the best bimodal particle-size distribution ($\lambda_{53}\approx 7$, $\zeta_{53}=2$). However, the fixed-diameter ratio scheme ($\lambda_{\mathrm{cmf}}=\lambda_{\mathrm{cmf}}\approx 49:7:1$) produced better results compared with the fixed-mass ratio scheme ($\zeta_{\mathrm{cmf}}=2:1:0.5$). Combining this diameter ratio ($\lambda_{531}$) with the optimal mass ratio ($\zeta_{531}=2:1:0.3$), the corresponding apparent viscosity (when $\dot{\gamma }=1\;s^{-1}$ and ϕ=65 wt.%) was 28% less than that obtained from the best bimodal particle-size distribution.

For both bimodal and trimodal particle-size distributions, the relationship between the relative viscosity $\eta_{r}$ and the reduced solid content $\phi /\phi_{m}$ can be described by the equation proposed by Krieger and Dougherty (i.e., $\eta_{r}={\left(1-\phi /\phi_{m}\right)}^{-\alpha }$). Given the suspension composition and particle shape, the maximum solid content $\phi_{m}$ depended only on the diameter ratio λ and mass ratio ζ and was independent of the shear rate. In contrast, the exponent α decreased with the shear rate. When the shear rate was 1, 10, 100, or 1000 $s^{-1}$, α was 3.94, 3.33, 2.73, or 2.01, respectively. However, at a given shear rate, all the $\eta_{r}$-$\phi /\phi_{m}$ rheological data of the DNP/HMX melt-case explosive suspension collapsed to a single curve, regardless of whether the particle-size distribution was bimodal or trimodal.

However, the best diameter ratio λ and mass ratio ζ obtained from both bimodal and trimodal particle-size distributions were local optima, not global optima, although we compared all possible combinations of particle-size distributions. Moreover, the above equation correlating $\eta_{r}$ with $\phi /\phi_{m}$ is essentially semi-empirical, and thus its generalization to DNP/HMX melt-cast explosive suspensions requires further investigation.

## Figures and Tables

**Figure 1 polymers-15-01446-f001:**
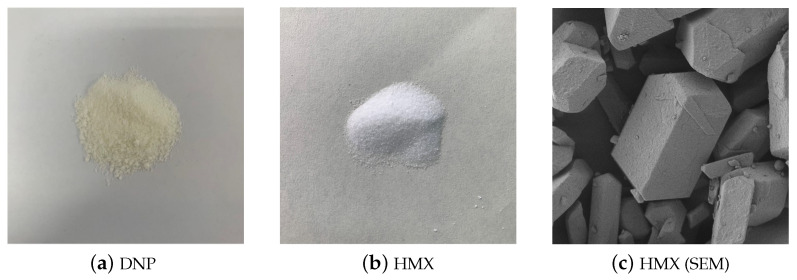
Photographs of DNP and HMX samples.

**Figure 2 polymers-15-01446-f002:**
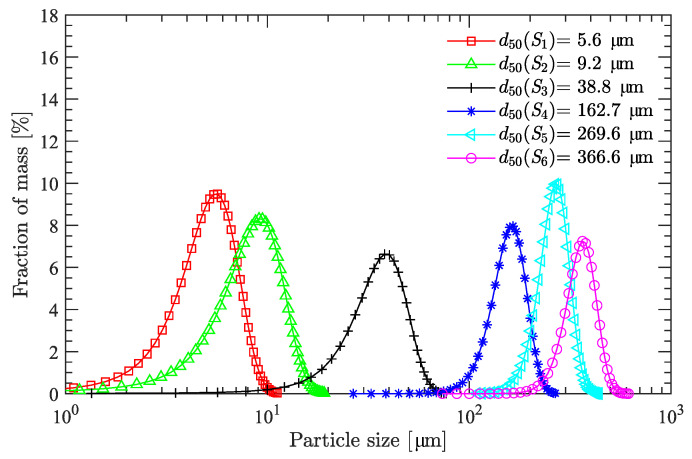
Particle-size distributions of six HMX samples.

**Figure 3 polymers-15-01446-f003:**
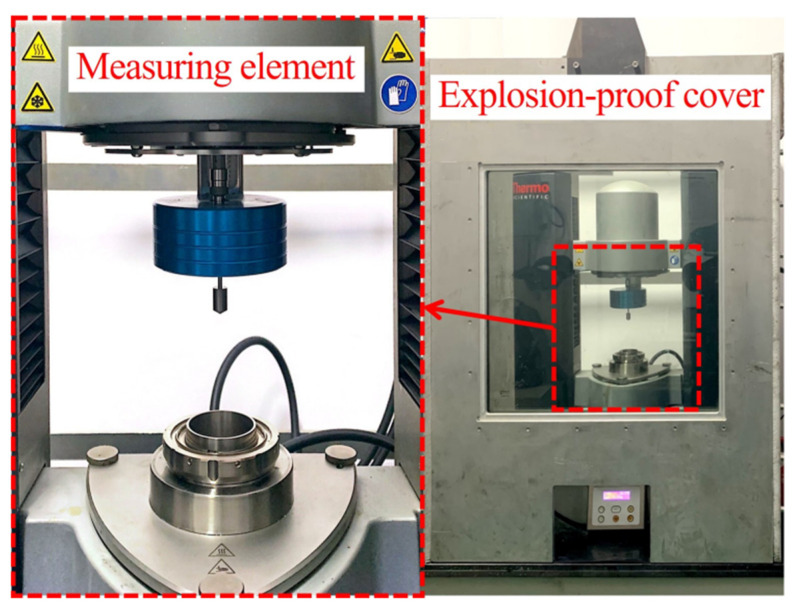
Photograph of the viscosity-measurement setup.

**Figure 4 polymers-15-01446-f004:**
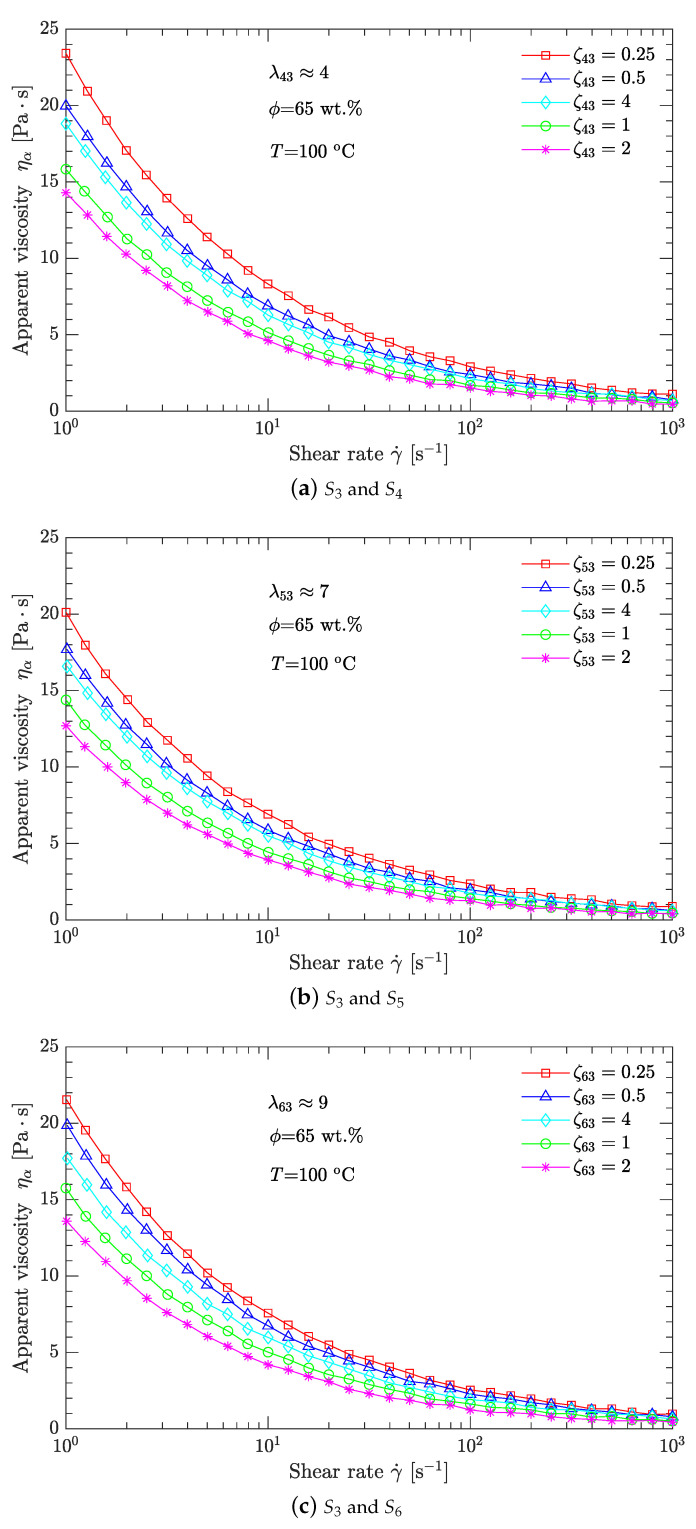
Effects of the shear rate $\dot{\gamma }$ on the apparent viscosity $\eta_{a}$ of DNP/HMX melt-cast explosive suspensions.

**Figure 5 polymers-15-01446-f005:**
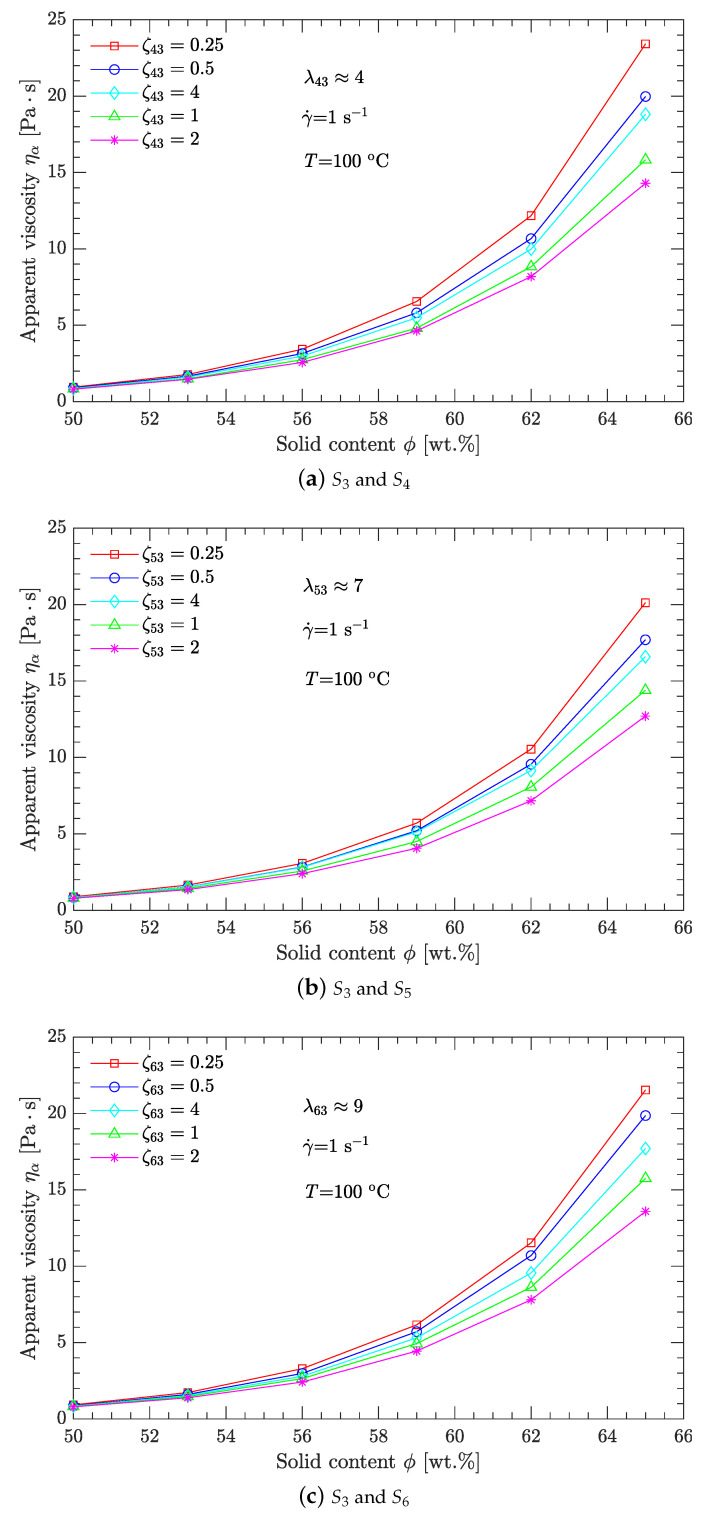
Effects of the solid content ϕ on apparent viscosity $\eta_{a}$ of the DNP/HMX melt-cast explosive suspensions.

**Figure 6 polymers-15-01446-f006:**
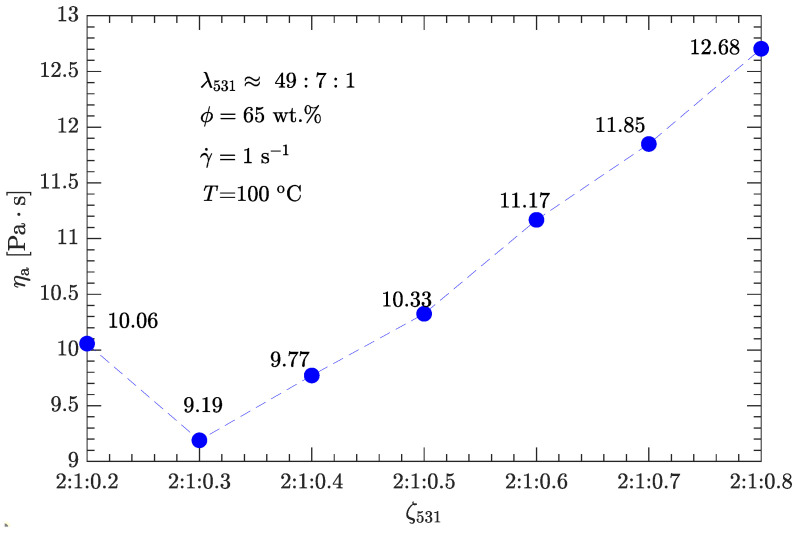
Effects of mass ratio $\zeta_{531}$ on apparent viscosity $\eta_{a}$ of the DNP/HMX melt-cast explosive suspensions.

**Figure 7 polymers-15-01446-f007:**
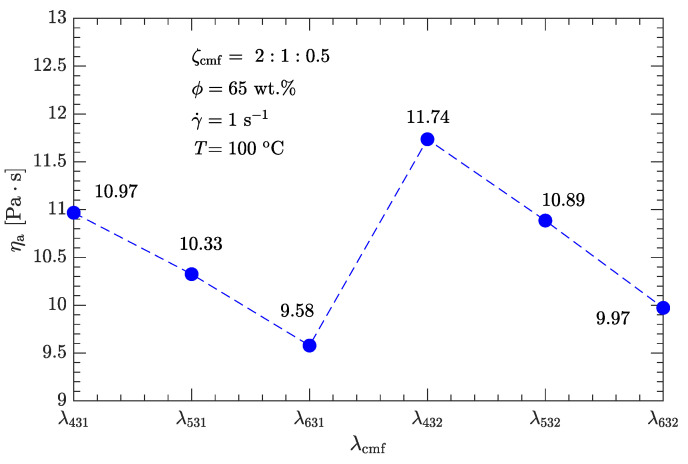
Effects of diameter ratio $\lambda_{\mathrm{cmf}}$ on apparent viscosity $\eta_{a}$ of DNP/HMX melt-cast explosive suspensions.

**Figure 8 polymers-15-01446-f008:**
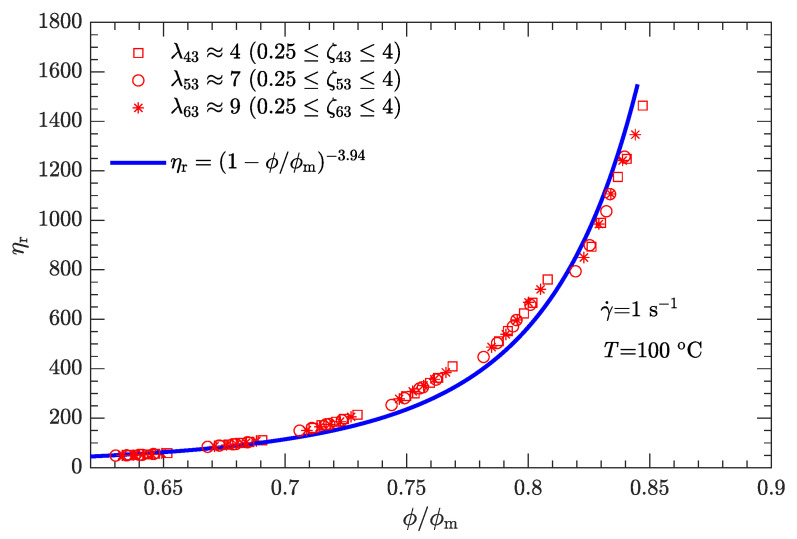
The relative viscosity $\eta_{r}$ as a function of reduced solid content $\phi /\phi_{m}$ for bimodal particle-size distributions when $\dot{\gamma }=1$$s^{-1}$.

**Figure 9 polymers-15-01446-f009:**
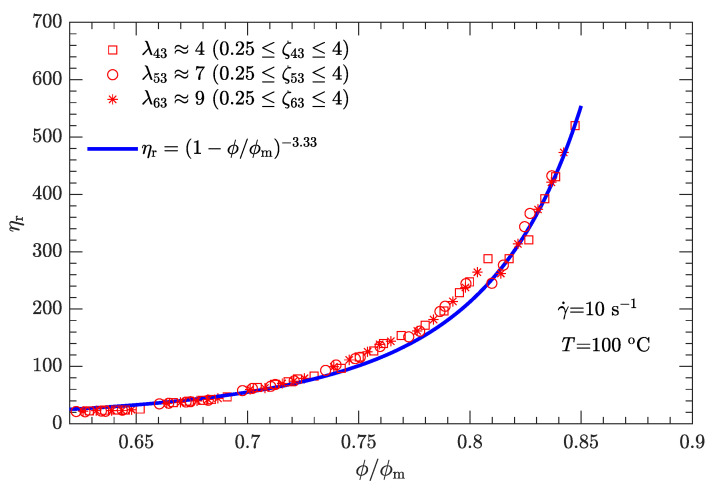
Relative viscosity $\eta_{r}$ as a function of reduced solid content $\phi /\phi_{m}$ for bimodal particle-size distributions with $\dot{\gamma }=10\;s^{-1}$.

**Figure 10 polymers-15-01446-f010:**
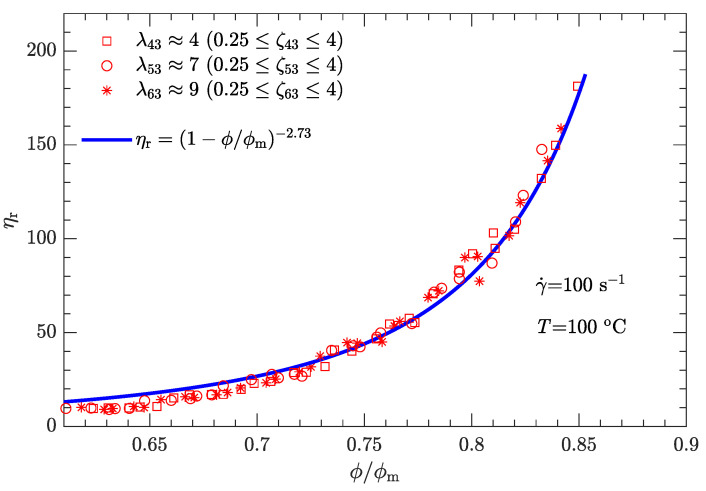
Relative viscosity $\eta_{r}$ as a function of reduced solid content $\phi /\phi_{m}$ for bimodal particle-size distributions with $\dot{\gamma }=100\;s^{-1}$.

**Figure 11 polymers-15-01446-f011:**
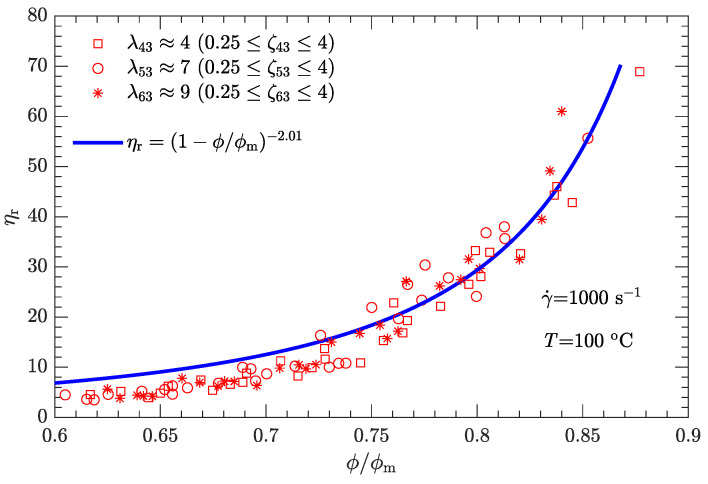
Relative viscosity $\eta_{r}$ as a function of reduced solid content $\phi /\phi_{m}$ for bimodal particle-size distributions with $\dot{\gamma }=1000\;s^{-1}$.

**Figure 12 polymers-15-01446-f012:**
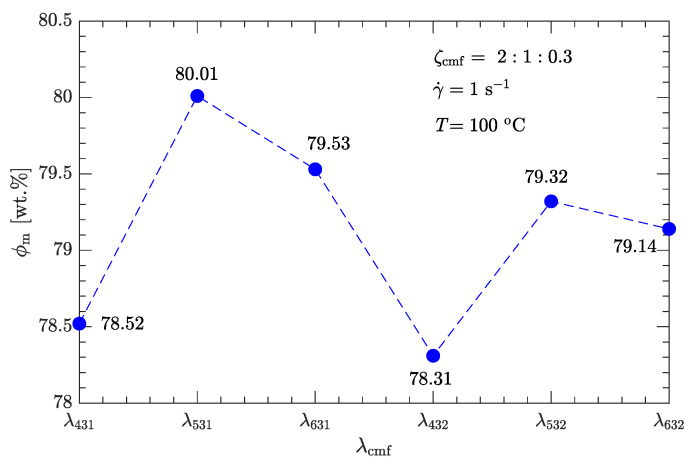
Effects of diameter ratio $\lambda_{\mathrm{cmf}}$ on maximum solid content $\phi_{m}$ of DNP/HMX melt-cast explosive suspensions.

**Figure 13 polymers-15-01446-f013:**
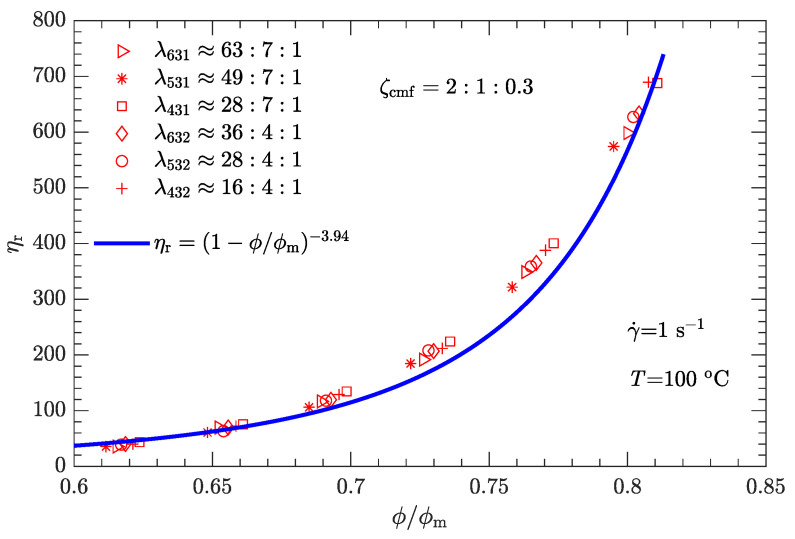
Relative viscosity $\eta_{r}$ as a function of reduced solid content $\phi /\phi_{m}$ for trimodal particle-size distributions with $\dot{\gamma }=1\;s^{-1}$.

**Figure 14 polymers-15-01446-f014:**
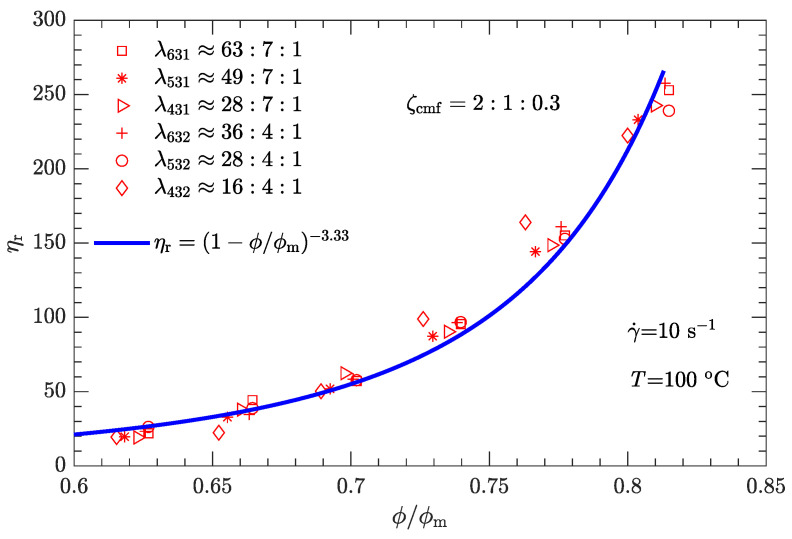
Relative viscosity $\eta_{r}$ as a function of reduced solid content $\phi /\phi_{m}$ for trimodal particle-size distributions with $\dot{\gamma }=10\;s^{-1}$.

**Figure 15 polymers-15-01446-f015:**
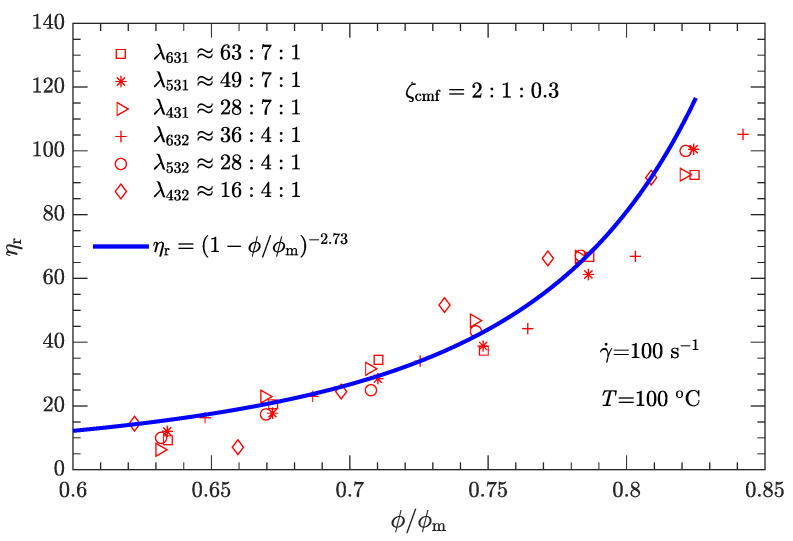
Relative viscosity $\eta_{r}$ as a function of reduced solid content $\phi /\phi_{m}$ for trimodal particle-size distributions with $\dot{\gamma }=100\;s^{-1}$.

**Figure 16 polymers-15-01446-f016:**
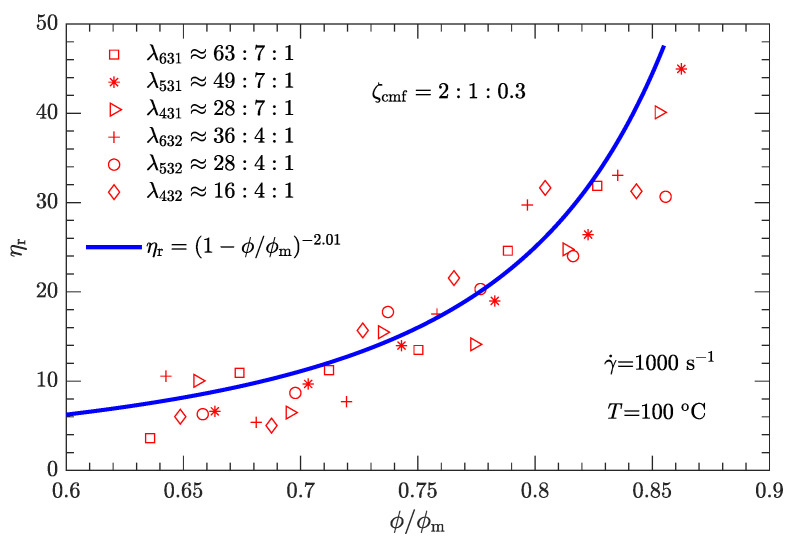
Relative viscosity $\eta_{r}$ as a function of reduced solid content $\phi /\phi_{m}$ for trimodal particle-size distributions with $\dot{\gamma }=1000\;s^{-1}$.

**Table 1 polymers-15-01446-t001:** Schemes of particle gradation for bimodal size distribution.

Scheme No.	$\lambda_{43}$	$\zeta_{43}$	Scheme No.	$\lambda_{53}$	$\zeta_{53}$	Scheme No.	$\lambda_{63}$	$\zeta_{63}$
1	4	0.25	6	7	0.25	11	9	0.25
2		0.5	7		0.5	12		0.5
3		1	8		1	13		1
4		2	9		2	14		2
5		4	10		4	15		4

**Table 2 polymers-15-01446-t002:** Apparent viscosities of DNP/HMX explosive suspensions with solid content 65 wt.%.

Diameter Ratio	Mass Ratio	Apparent Viscosity $\eta_{a}$ [Pa·s]			
		$\dot{\gamma }=1$ $s^{-1}$	$\dot{\gamma }=10$ $s^{-1}$	$\dot{\gamma }=100$ $s^{-1}$	$\dot{\gamma }=1000$ $s^{-1}$
$\lambda_{43}\approx 4$	$\zeta_{43}=2$	14.29	4.61	1.52	0.45
$\lambda_{63}\approx 9$	$\zeta_{63}=2$	13.59	4.19	1.24	0.43
$\lambda_{53}\approx 7$	$\zeta_{53}=2$	12.70	3.92	1.22	0.39

**Table 3 polymers-15-01446-t003:** Apparent viscosities of the DPN/HMX explosive suspensions at a shear rate of 1 $s^{-1}$.

Diameter Ratio	Mass Ratio	Apparent Viscosity $\eta_{a}$ [Pa·s]					
		ϕ=50 [wt.%]	ϕ=53 [wt.%]	ϕ=56 [wt.%]	ϕ=59 [wt.%]	ϕ=62 [wt.%]	ϕ=65 [wt.%]
$\lambda_{43}\approx 4$	$\zeta_{43}=2$	0.81	1.46	2.56	4.63	8.18	14.29
$\lambda_{63}\approx 9$	$\zeta_{63}=2$	0.79	1.39	2.42	4.45	7.80	13.59
$\lambda_{53}\approx 7$	$\zeta_{53}=2$	0.78	1.35	2.40	4.06	7.16	12.70

**Table 4 polymers-15-01446-t004:** Schemes of particle gradation for trimodal size distribution with a fixed diameter ratio.

Scheme No.	Diameter Ratio $\lambda_{531}$	Mass Ratio $\zeta_{531}$
16	49:7:1	2:1:0.2
17		2:1:0.3
18		2:1:0.4
19		2:1:0.5
20		2:1:0.6
21		2:1:0.7
22		2:1:0.8

**Table 5 polymers-15-01446-t005:** Schemes of particle gradation for trimodal size distribution with a fixed mass ratio.

Scheme No.	Diameter Ratio	Mass Ratio
23	$\lambda_{431}\approx 28:7:1$	2 : 1 : 0.5
19	$\lambda_{531}\approx 49:7:1$	
24	$\lambda_{631}\approx 63:7:1$	
25	$\lambda_{432}\approx 16:4:1$	
26	$\lambda_{532}\approx 28:4:1$	
27	$\lambda_{632}\approx 36:4:1$	

**Table 6 polymers-15-01446-t006:** Effects of the diameter ratio and mass ratio on the maximum solid content for bimodal particle-size distributions.

Scheme No.	Diameter Ratio	Mass Ratio	[$\eta_{a}$ Pa·s]	$\phi_{m}$ [wt.%]
9	$\lambda_{53}\approx 7$	$\zeta_{53}=2$	12.70	77.73
14	$\lambda_{63}\approx 9$	$\zeta_{63}=2$	13.59	77.31
4	$\lambda_{43}\approx 4$	$\zeta_{43}=2$	14.29	77.02
8	$\lambda_{53}\approx 7$	$\zeta_{53}=1$	14.39	76.93
13	$\lambda_{63}\approx 9$	$\zeta_{63}=1$	15.76	76.81
3	$\lambda_{43}\approx 4$	$\zeta_{43}=1$	15.83	76.70
10	$\lambda_{53}\approx 7$	$\zeta_{53}=4$	16.58	76.53
7	$\lambda_{53}\approx 7$	$\zeta_{53}=0.5$	17.69	76.32
15	$\lambda_{63}\approx 9$	$\zeta_{63}=4$	17.71	76.23
5	$\lambda_{43}\approx 4$	$\zeta_{43}=4$	18.81	76.01
12	$\lambda_{63}\approx 9$	$\zeta_{63}=0.5$	19.86	75.89
2	$\lambda_{43}\approx 4$	$\zeta_{43}=0.5$	19.97	75.78
6	$\lambda_{53}\approx 7$	$\zeta_{53}=0.25$	20.12	75.72
11	$\lambda_{63}\approx 9$	$\zeta_{63}=0.25$	21.54	75.41
1	$\lambda_{43}\approx 4$	$\zeta_{43}=0.25$	23.42	75.12

**Table 7 polymers-15-01446-t007:** Schemes of particle gradation for trimodal size distribution with the optimal mass ratio shown in Figure 6.

Scheme No.	Diameter Ratio	Mass Ratio
28	$\lambda_{431}\approx 28:7:1$	2 : 1 : 0.3
29	$\lambda_{531}\approx 49:7:1$	
30	$\lambda_{631}\approx 63:7:1$	
31	$\lambda_{432}\approx 16:4:1$	
32	$\lambda_{532}\approx 28:4:1$	
33	$\lambda_{632}\approx 36:4:1$	

## Data Availability

The data presented in this study are available on request from the corresponding author.

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
