# Peer review of "Rheological Behavior of DNP/HMX Melt-Cast Explosives with Bimodal and Trimodal Particle-Size Distributions"

_polymers, 2023, doi:10.3390/polym15061446_

Round 1

Reviewer 2 Report

It is a high-level paper describing in detail the optimization of the rheology of novel pyrazole-based composition. I have only minor recommendations:

1) it could be mentioned that the applied in other field additive manufacturing technologies are only emerging in energetic materials area [https://doi.org/10.1002/prep.201900060]. Hence the actuality of melt-casting, the topic of this report

2) during processing of the melt, the volatility is important. I'm not aware of the vapor pressure determination for 3,4-dinitropyrazole, but for most close chemical, 3,5- isomer the vapor pressure on the TNT level has been reported (https://doi.org/10.1016/j.tca.2020.178697)

3) "since the HMX particles used in the formulation of explosives generally measure 10^0–10^2 μm," - it is better to write simply 1-100 microns, and later in text

4) please give more clear results in the conclusions section: what optimal compositions are for bi- and trimodal powders (e.g., simply the concentration of solid HMX)

5) have you analyzed the long-term stability of HMX in molten DNP?

Round 2

Reviewer 1 Report

Please make sure to address all questions from the original review in your manuscript. Show the new sections of your revised manuscript in a different colors so that the changes are easy to see. Few improvements have been made as far as I can see.

Author Response

Dear reviewer: 
Thanks a lot for your valuable comments, and the corresponding changes to our manuscript are described below:

Comment 1: Figure 1 should have size scales. What is the difference between b and c?
Response:  The size scale was added in Fig. 1(c). Description of the difference between Figs. 1(b) and 1(c) were added in the 1st Paragraph of Section 2.1,
Figs. 1(b) and 1(c) show the photographs of HMX particles (sample S3) taken by camera and by scanning electron microscope (SEM), respectively. 

Comment 2: Are there issues with slip in the rheometer? Were roughened plates used to prevent slip?
Response: The Haake Mars III rheometer is well designed with consideration of slip prevention problem. Therefore, the possible wall slip effect was assumed to be negligible in this study. Some comments on wall slip were added in the 1st Paragraph of Section 2.2.1,
Moreover, wall slip is commonly observed in rheometry suspensions, and such phenomenon is generally associated to many factors, such as low shear rates and smooth walls of measuring geometries[1,2]. However, the specific analysis of the wall slip phenomenon will not be detailed in this study.

Comment 3: What geometry was the rheometer? Cone and Plate, Plate and plate, or Couette? I can't tell from the image, but I am glad it was designed to be explosion proof. High risk experiments!
Response: The Couette geometry was used in the rheometer. Description of this geometry was added in the 1st Paragraph of Section 2.2.1,
The measuring element has a Couette geometry with a gap of 2.0 mm.

Description of the suspension temperature was added in the 2nd Paragraph of Section 2.2.1,
(where an oil bath was used due to the risk of explosive samples)

Comment 4: Were there issues of particle migration in the rheometer? Are we sure the material stays well mixed and is in a viscometric flow?
Response: Particle migration phenomenon can not be completely eliminated in the rheometer with a Couette geometry. Comments on this phenomenon were added in the 1st Paragraph of Section 2.2.1,
Furthermore, the inner and outer radii of the Couette geometry were 8.5 mm and 10.5 mm, respectively. Therefore, the assumption of constant shear rate in the gap may be well respected, and the particle migration was assumed to be negligible, and the material be well mixed.
Two references were added in the 1st Paragraph of Section 2.2.1.

Comment 5: Is the Krieger model the best choice for the rheological model, especially since we see a shear rate dependence of the viscosity? Also, the Krieger model has an exponent of . Where [ŋ] is the intrinsic viscosity and is the maximum packing. The Krieger model is written in terms of only particle volume fraction and was specifically developed for monomodal particles. It is not widely used for bimodal and trimodal particle distributions. They incorporate the polydispersity in particle size by using a higher value of the maximum packing. They incorporate shear rate by varying the exponent of the Krieger viscosity as function of the applied shear rate. This seems like “off‐road” usage of the model and doesn't seem well justified other than that it kind of fits the data. I also worry that the shear‐rate dependence of the viscosity measurements could be due to particle migration.
Response: The Krieger model is widely used to describe the relationship between relative viscosity and reduced solid content of suspensions. Moreover, on the basis of the data shown in Fig. 4, the DNP/HMX melt-cast explosive suspension presents a non-Newtonian (shear thinning) viscous behavior, thus the relative viscosity of this explosive suspensions must be a function of shear rate. Therefore, it is reasonable to vary the exponent of the Krieger viscosity as function of the applied shear rate, since physically the maximum solid content is dominated by particle shape and particle-size distribution and assumed to have nothing to do with the shear rate.

Comment 6: Do HMX and DNP have the same density? The Krieger model should be expressed in terms of volume fraction, not weight fraction.
Response: The densities of solid HMX and molten DNP are 1.91 g/cm^3 and 1.74 g/cm^3, respectively. When modeling the viscosity versus concentration relationship for general suspensions, volume fraction is widely used in the model. However, for explosive suspensions, weight fraction or solid content is more often used by both academic and industrial communities of energetic materials.

Comment 7:There is so much data, especially for the trimodal case, that it gets difficult to digest what the findings are. Maybe give fewer cases? Add some images of the various microstructures so that the reader can follow the design of experiments?
Response: When minimizing the viscosity of suspensions with bimodal or trimodal particle-size distributions, there are a lot of combinations of different diameter ratios and mass ratios. One of the most important findings of this study is that the optimal values of diameter ratio and mass ratio for trimodal case can be directly constructed on the basis of those values for bimodal case.

Adding some images of the various microstructures with bimodal or trimodal particle-size distribution is a great idea. However, we have not taken such photographs. Maybe we can do that in our future investigations.

Comment 8: It might be better off starting with a shear thinning model, like a Carreau fluid, and adding particle concentration to that?
Response: When modeling the relative viscosity versus reduced solid content relationship for shear thinning suspensions, the Krieger model is widely used. However, the Carreau or Power law model maybe an alternative choice if particle concentration is effectively embedded in this model. We hope such valuable suggestion will be implemented in our future study.

Comment 9: “In addition to rheological modifications, bidisperse suspensions in nonuniform shear such as pressure‐driven flow, where particle migration occurs, are known to display segregation behavior based on particle size [28,29], but here we will focus on conditions where the particle sizes remain well‐mixed.” [2]. Thus, if we don’t have a constant shear rate flow in the rheometer, we can get particle migration as well as size segregation.
Response: The DNP/HMX explosive suspension has been well stirred to ensure uniform mixing and to eliminate solid agglomerates. Moreover, the ratio of the outer and inner radii of the Couette geometry is relatively small, and the gap is also small. Therefore, the assumption of constant shear rate in the gap may be well respected, and both the particle migration and size segregation were assumed to be negligible in this study.

References

[1] Fernandes, R.; Turezo, G.; Andrade, D.; Franco, A.; Negrão, C. Are the rheological properties of water-based and synthetic drilling fluids obtained by the Fann 35A viscometer reliable? Journal of Petroleum Science and Engineering 2019, 177, 872–879.

[2]Talmon, A.; Meshkati, E., Rheology, Rheometry and Wall Slip; 2022.
